# Assessing and Modelling of Post-Traumatic Stress Disorder Using Molecular and Functional Biomarkers

**DOI:** 10.3390/biology12081050

**Published:** 2023-07-26

**Authors:** Konstantina Skolariki, Aristidis G. Vrahatis, Marios G. Krokidis, Themis P. Exarchos, Panagiotis Vlamos

**Affiliations:** Bioinformatics and Human Electrophysiology Laboratory, Department of Informatics, Ionian University, 49100 Corfu, Greece; c19skol@ionio.gr (K.S.); aris.vrahatis@ionio.gr (A.G.V.); exarchos@ionio.gr (T.P.E.); vlamos@ionio.gr (P.V.)

**Keywords:** post-traumatic stress disorder, biomarkers, neurotransmitter imbalance, hypothalamic-pituitary-adrenal axis, network analysis

## Abstract

**Simple Summary:**

Current global statistics indicate that approximately 1 in 13 individuals will develop post-traumatic stress disorder (PTSD) at some point in their lives. PTSD is a complex psychological disorder that emerges as a result of exposure to traumatic events, and its intricate nature poses challenges for diagnosis. The focus should be directed towards identifying the catalytic factors of PTSD, which can serve as molecular or functional biomarkers. To address this, an extensive review of research sources to compile a comprehensive list of catalytic factors at both the molecular and functional levels was conducted. A mathematical model encompassing all the significant catalytic factors was proposed, with the future aim of developing a holistic model of PTSD that will incorporate a panel of characteristics. Furthermore, the behavior of these factors at the genetic level, considering both ontological and network perspectives was explored, enabling the gain of a deeper understanding of PTSD by capturing their interconnectivity and biological roles. To the best of our knowledge, this study represents the first comprehensive examination and modeling of catalytic factors in PTSD, taking into account their interconnected associations.

**Abstract:**

Post-traumatic stress disorder (PTSD) is a complex psychological disorder that develops following exposure to traumatic events. PTSD is influenced by catalytic factors such as dysregulated hypothalamic-pituitary-adrenal (HPA) axis, neurotransmitter imbalances, and oxidative stress. Genetic variations may act as important catalysts, impacting neurochemical signaling, synaptic plasticity, and stress response systems. Understanding the intricate gene networks and their interactions is vital for comprehending the underlying mechanisms of PTSD. Focusing on the catalytic factors of PTSD is essential because they provide valuable insights into the underlying mechanisms of the disorder. By understanding these factors and their interplay, researchers may uncover potential targets for interventions and therapies, leading to more effective and personalized treatments for individuals with PTSD. The aforementioned gene networks, composed of specific genes associated with the disorder, provide a comprehensive view of the molecular pathways and regulatory mechanisms involved in PTSD. Through this study valuable insights into the disorder’s underlying mechanisms and opening avenues for effective treatments, personalized interventions, and the development of biomarkers for early detection and monitoring are provided.

## 1. Introduction

Post-traumatic stress disorder (PTSD) is a multifaceted psychological disorder that may arise following exposure to traumatic events. The etiology and progression of PTSD involve a multitude of genetic and neurochemical factors that interact in complex ways. Gaining a comprehensive understanding of these underlying mechanisms and factors influencing the onset and progression of PTSD is crucial for developing effective prevention and treatment strategies [1]. Catalytic factors play a significant role in shaping the progression and severity of PTSD and its symptoms. The catalytic factors associated with PTSD include dysregulation of the hypothalamic-pituitary-adrenal (HPA) axis [2], which leads to altered glucocorticoid levels. Neurotransmitter imbalances [3], such as disruptions in serotonin, dopamine, and norepinephrine, also contribute to PTSD development and progression. Additionally, oxidative stress, characterized by an imbalance between reactive oxygen species and antioxidant defenses, plays a role in the pathophysiology of PTSD [4]. Gene variations can also act as catalytic factors and play a significant role in the development and progression of PTSD. These genetic variations can modulate important biological processes and pathways, leading to alterations in neurochemical signaling, synaptic plasticity, and stress response systems. Understanding the role of catalytic factors in PTSD can lead to valuable insights regarding the heterogeneity of symptom presentation and treatment response among individuals affected by PTSD. Furthermore, while information regarding the catalytic factors of PTSD, particularly in the form of genes, exists at the individual level, further analyses conducted from a network perspective can explore the connectivity and associations among these molecular factors. Examining their roles within various ontologies allows for the assessment of their potential involvement and behavior in diverse molecular processes. The mapping of these factors in biological networks holds the potential to provide a deeper understanding, ultimately leading to the discovery of novel insights into the development and manifestation of PTSD.

## 2. Catalytic Factors Associated with PTSD

There are several catalytic factors that can affect PTSD development and progression. These factors include dysregulation of the HPA axis, changes in glucocorticoid levels, imbalances in neurotransmitters, inflammatory markers [5], epigenetic changes [6], and genetic variations [7] in genes related to stress response and resilience. These factors can contribute to the complex interplay of biological processes involved in PTSD. Regarding genetic variations, there is evidence suggesting that they may play a role in the development of PTSD. Genetic variations can contribute to individual differences in vulnerability and resilience to traumatic experiences. Alterations in DNA repair pathways have been proposed as a potential mechanism through which genetic variations can increase susceptibility to mutagenicity and genomic instability [8]. Mutagenicity refers to the ability of a substance to induce mutations in the DNA of an organism. The identified genetic variations associated with PTSD are shown in Table 1. It’s important to note that genetic factors alone are unlikely to fully account for the development of PTSD, as the condition is influenced by a complex interplay of genetic, environmental, and psychological factors.

### 2.1. Oxidative Stress

Growing evidence indicates that oxidative stress is implicated in the development and persistence of PTSD, highlighting the importance of the imbalance between reactive oxygen species (ROS) production and antioxidant defense mechanisms [9,10,11]. Several biomarkers of oxidative stress have been identified in association with PTSD, offering insights into its pathophysiology and potential diagnostic and prognostic applications [9,10,11]. One such biomarker is malondialdehyde (MDA), which has shown elevated levels in individuals with PTSD, particularly in relation to hyperarousal symptoms [9]. This suggests that increased MDA may contribute to the severity and persistence of PTSD symptoms. Paraoxonase 1 (PON1), although not a direct marker of oxidative stress, has been found to be inversely correlated with oxidative stress markers, suggesting that reduced PON1 activity may indicate heightened oxidative stress levels [9]. Measuring PON1 activity can serve as an indirect indicator of oxidative stress, potentially aiding in the assessment of PTSD and its associated oxidative burden. Furthermore, studies have demonstrated decreased levels of glutathione (GSH), an essential antioxidant, in individuals with PTSD compared to controls [10]. This reduction in GSH suggests compromised antioxidant defense mechanisms in individuals with PTSD, contributing to increased vulnerability to oxidative damage. Similarly, decreased activity of superoxide dismutase (SOD), glutathione peroxidase (GPX), and catalase (CAT) has been observed in PTSD patients, further supporting the involvement of oxidative stress in the disorder [11]. Taken together, these findings highlight the potential role of oxidative stress in the etiology and persistence of PTSD. Biomarkers such as MDA, PON1, GSH, SOD, GPX, and CAT offer valuable insights into the oxidative imbalance associated with PTSD and hold promise as diagnostic and prognostic tools. Considering, the fact that PTSD is a complex and multifactorial disorder, a panel of biomarkers would work best as such a tool.

### 2.2. Genes Variations Related to HPA Axis

The HPA axis is a complex network of interactions between the hypothalamus, pituitary gland, and adrenal gland that is responsible for regulating the body’s response to stress. In individuals with PTSD, the HPA axis is often dysregulated. This can lead to changes in the levels of glucocorticoids (GCs), which are hormones that play a role in regulating the immune system, metabolism, and the body’s response to stress. The dysregulation of the HPA axis and changes in glucocorticoid levels may contribute to the development and maintenance of PTSD symptoms, such as hyperarousal, avoidance, and re-experiencing. Additionally, changes in the HPA axis have been linked to alterations in the immune system, inflammation, and changes in neurotransmitter levels, which may also contribute to the development and progression of PTSD [2]. The activity of the HPA axis in individuals with PTSD can vary, and it is not always consistently hyperactive or hypoactive. It is worth noting that individual differences, comorbidities, and the timing of cortisol measurements can influence the observed HPA axis activity in PTSD. Factors like chronicity of symptoms, time since trauma exposure, and the presence of comorbid conditions, such as depression or anxiety, can all contribute to the variability of HPA axis functioning in individuals with PTSD. The case of HPA axis hyperactivation is characterized by increased release of corticotropin-releasing factor (CRF) which is typically associated with acute stress responses. However, the dysregulation of the HPA axis that can occur in PTSD may result in alterations in CRF levels [12]. The HPA axis begins with the release of corticotropin-releasing hormone (CRH) from the hypothalamus in response to various stressors or triggers. CRH stimulates the release of adrenocorticotropic hormone (ACTH). A study showed that ACTH levels were higher in PTSD [13]. This suggests an imbalance in the HPA axis activity, with an elevated release of ACTH in individuals with PTSD.

In PTSD, there is evidence of dysregulated glucocorticoid signaling, referring to the interactions and effects of GCs on various physiological processes. GCs also play a role in modulating inflammatory processes. In individuals with PTSD, alterations in GC signaling can contribute to dysregulated immune responses and inflammation. Epigenetic modifications, including DNA methylation and histone modifications, can influence GC signaling. In individuals with PTSD, alterations in epigenetic regulation of GR gene expression and GC signaling pathway genes have been reported as analyzed below. These epigenetic changes can impact the sensitivity and functioning of the GR, contributing to GC signaling dysregulation in PTSD. It’s important to note that the dysregulation of GC signaling in PTSD is complex and can vary among individuals. Further research is needed to elucidate the precise mechanisms involved in GC signaling dysregulation in PTSD and its implications for the development and maintenance of the disorder [14]. Alterations in glucocorticoid receptors (GR) have also been noted. Research suggests that individuals with PTSD may exhibit enhanced sensitivity of the GR, leading to increased negative feedback inhibition of cortisol release in the HPA axis. This enhanced GR sensitivity could contribute to alterations in stress response regulation observed in PTSD [15]. Cortisol levels in PTSD can also vary depending on specific subtypes of PTSD and the presence of comorbidities. Comorbid conditions such as depression or anxiety may influence cortisol levels in individuals with PTSD. It is important to note that there is considerable variability in cortisol levels among individuals with PTSD, and not all individuals will exhibit the same cortisol patterns. The timing of cortisol measurements, chronicity of symptoms, and other individual factors can contribute to these variations. One hallmark feature of PTSD is an exaggerated or dysregulated cortisol response to stress. Some individuals with PTSD may exhibit an increased cortisol response to stressors, which can indicate hyperactivity of the HPA axis. On the other hand, there are also studies suggesting blunted or reduced cortisol responses to stress in some individuals with PTSD, indicating HPA axis hypoactivity. This dysregulation can be a result of the trauma itself or other factors such as genetics and epigenetics [16]. Overall, the dysregulation of the HPA axis in PTSD is thought to contribute to the persistent and chronic nature of the disorder.

In PTSD, alterations in the HPA axis can impact the effectiveness of standard first-line treatments like trauma-focused cognitive behavioral therapy (TF-CBT) due to the importance of cortisol in memory consolidation and retrieval. The lack of cortisol can hinder the extinction of fear responses associated with traumatic memories, necessitating alternative treatments that target the HPA axis and regulate cortisol levels for non-responsive individuals [17]. The Nuclear Receptor Subfamily 3 Group C Member 1 (NR3C1) gene plays a role in PTSD as genetic variations have been associated with altered glucocorticoid receptor function, leading to HPA axis dysregulation and impairments in stress response and emotion regulation. Several SNPs in the NR3C1 gene, such as Bcl-1 and ER22/23EK, have been identified as potential risk factors for PTSD. These SNPs are associated with reduced glucocorticoid sensitivity and increased vulnerability to developing PTSD following trauma exposure. Homozygous carriers of the Bcl-1 polymorphism (rs41423247) have been shown to have a higher risk of major depression and PTSD, as well as increased rates of suicide [18,19,20]. The 9β SNP variation in NR3C1 has also been linked to altered glucocorticoid sensitivity and investigated in relation to PTSD [21]. Additionally, the rs258747 SNP in NR3C1 has been associated with PTSD [22].

The FK506 Binding Protein 5 (FKBP5) gene has also been implicated in PTSD. Variants of FKBP5 have been found to significantly moderate the effects of early-life stress on PTSD. Several FKBP5 SNPs, including rs3800373, rs9296158, rs1360780, and rs9470080, have been associated with lifetime probable PTSD [22,23]. Another HPA axis-associated genetic variation involves the Corticotropin Releasing Hormone Receptor 1 (CRHR1) gene. CRHR1 has been linked to an increased risk for PTSD symptoms. The rs110402 variation in CRHR1 has been associated with an elevated risk of PTSD symptoms, and other SNPs such as rs12938031, rs12944712, and rs4792887 have been associated with PTSD symptoms and diagnosis [24,25]. In a similar framework, the Corticotropin Releasing Hormone Receptor 2 (CRHR2) gene encodes a protein involved in coordinating physiological and behavioral responses to stress. In a study of HPA axis genes and their association with PTSD symptoms in Chinese earthquake survivors, a specific SNP in CRHR2, rs2267715, was found to have main effects associated with PTSD severity and most PTSD symptom clusters except dysphoric arousal. This suggests that variations in CRHR2 may contribute to the development and severity of PTSD symptoms [26].

### 2.3. Neurotransmitter Imbalances and Related Genes Variations

Neurotransmitter imbalances play a crucial role in the development and progression of PTSD, as evidenced by numerous studies. Serotonin, a neurotransmitter involved in mood regulation, anxiety, and stress, has been associated with PTSD and other anxiety disorders [27]. Similarly, dysregulation in norepinephrine, a neurotransmitter involved in the body’s “fight or flight” response, has been linked to hyperarousal symptoms in PTSD [28]. Elevated norepinephrine levels have been observed in individuals with PTSD, contributing to their heightened state of arousal.

Research evidence suggests that low levels of serotonin contribute to the manifestation of PTSD symptoms. More precisely, dysregulation of the brain’s serotonergic system is implicated in the pathophysiology of PTSD, with serotonin playing a vital role in regulating emotional responses and overall emotionality [29]. Genetic variations in serotonergic system genes have been identified as potential contributors to the development of PTSD. The solute carrier family 6 member 4 (SLC6A4) gene is of particular interest in the context of PTSD. One polymorphism within its promoter region is the serotonin transporter-linked polymorphic region (5-HTTLPR), where the short (S) allele is associated with reduced expression and function of the transporter compared to the long (L) allele. The S-allele’s reduced expression and function lead to decreased serotonin uptake, which can result in emotional instability and increase susceptibility to anxiety and depression. Additionally, the rs25531 single nucleotide polymorphism (SNP) located within the 5-HTTLPR region interacts with these effects. Due to their proximity and potential interaction in influencing serotonin transporter expression and function, these two polymorphisms are often analyzed together as a combined genotype [30]. The 5-hydroxytryptamine receptor 2A (HTR2A) gene is another gene of interest in relation to PTSD. The HTR2A SNP, rs7997012, interacts with PTSD severity to predict reduced connectivity within components of the brain’s default mode network (DMN), including the posterior cingulate cortex (PCC), right medial prefrontal cortex, and right middle temporal gyrus. Additionally, two other HTR2A SNPs, rs977003 and rs7322347, have been found to moderate the association between PTSD severity and the PCC-right MTG component of the DMN [31]. The Tryptophan hydroxylase 2 (TPH2) gene encodes the enzyme responsible for the initial and rate-limiting step in serotonin biosynthesis, a neurotransmitter implicated in mood regulation. Notably, the TPH2 rs11178997T allele has shown a significant association with DSM-5-based PTSD severity scores [32]. These findings highlight the involvement of genetic variations in the serotonergic system in the pathogenesis of PTSD.

Imbalances in other neurotransmitters, such as glutamate, GABA, and acetylcholine, have also been implicated in PTSD. Glutamate, an excitatory neurotransmitter, has shown dysregulation in brain regions like the anterior cingulate cortex and hippocampus in individuals with PTSD [33]. The dysregulation of glutamate transmission in these regions may contribute to the cognitive and emotional disturbances observed in PTSD. GABA, the primary inhibitory neurotransmitter involved in anxiety and stress regulation, exhibits reduced tone in individuals with PTSD [34]. This reduction in GABAergic overall activity and functioning in the brain is associated with increased anxiety and fear responses, contributing to the development and maintenance of PTSD symptoms. Studies have found lower levels of GABA in brain regions such as the prefrontal cortex and amygdala, which are involved in emotional processing and regulation. Furthermore, genetic variations in GABA receptor subunit genes have been linked to an increased risk for PTSD, highlighting the role of GABA in the disorder [34]. Dopamine, a neurotransmitter involved in reward processing and motivation, has also been linked to the development of PTSD [35]. The GABA system, which is involved in the regulation of neurotransmission and various physiological and neurological processes, has been implicated in the modulation of anxiety, stress, and fear responses, making it relevant to the study of PTSD. Dysregulation of the GABA system has been observed in individuals with PTSD, suggesting its involvement in the pathophysiology of the disorder. One specific gene that has been studied in relation to PTSD is the GABAA receptor subunit alpha 2 (GABRA2) gene. Polymorphisms in the GABRA2 gene have been investigated for their association with PTSD. A study found that three specific polymorphisms in GABRA2 had significant interactions with childhood trauma in predicting the development of PTSD. This suggests that genetic variations in GABRA2 may contribute to an individual’s vulnerability to PTSD, particularly in those who have experienced childhood trauma [36].

The Spindle and Kinetochore-associated Complex Subunit 2 (SKA2) gene and its rs7208505 polymorphism have been implicated in the development of PTSD. The SKA2 gene is involved in regulating the stress response and has been associated with an increased risk for depression and suicide. Individuals with PTSD have been found to exhibit lower levels of SKA2 gene expression compared to those without PTSD. Additionally, the rs7208505 polymorphism has been associated with lower SKA2 gene expression and higher PTSD symptoms, particularly in individuals with a history of childhood trauma [37]. While the Cannabinoid-1 receptor (CNR1) gene is not typically considered a candidate gene for the hypothalamic-pituitary-adrenal (HPA) axis, it is part of the endocannabinoid system, which may interact with the HPA axis and play a role in stress response and related disorders such as PTSD. Some evidence suggests that CNR1 genetic variations may be associated with HPA axis dysregulation and increased risk for PTSD. Another study investigated the effects of the G1359A genetic variant (rs1049353) in the CNR1 gene and childhood abuse on the development and expression of PTSD symptoms [38]. Another CNR1 variation associated with PTSD is the rs806371 [39]. The study found a trending association between the genotype at rs806371 in CNR1 and PTSD symptom severity at follow-up. It is important to note that further research is needed to fully understand the mechanisms by which the SKA2 and CNR1 genes contribute to PTSD and the specific roles of their genetic variations in the development and progression of the disorder.

Dysregulation of dopamine signaling pathways may contribute to the emotional dysregulation and altered reward processing observed in individuals with PTSD. These findings collectively highlight the significance of neurotransmitter imbalances in the development and maintenance of PTSD symptoms. Dopaminergic dysregulation in PTSD appears to involve alterations in the brain’s reward system, which is modulated by dopamine. Studies have demonstrated that individuals with PTSD exhibit reduced reward processing compared to healthy controls, indicating potential dysregulation of the mesolimbic dopaminergic pathway responsible for reward processing and motivation [40]. Furthermore, the dopaminergic system has been found to interact with the dysregulated HPA axis, as discussed earlier [41]. The dopamine receptor D2 gene (DRD2) is of particular interest in the context of PTSD. Within this gene, a specific SNP known as rs1800497 has been identified, which has two alleles: T (A1) and C (A2). The DRD2/ANKK1-Taq1A variant (rs1800497) is located in the ANKK1 gene, adjacent to the dopamine D2 receptor gene (DRD2) on chromosome 11. This polymorphism has been associated with the regulation of dopamine synthesis and reduced D2 receptor expression in the brain, which forms the pathophysiological basis for various PTSD symptoms. The rs1800497 polymorphism has been shown to contribute to the severity of PTSD symptoms, and individuals with the A1 allele are more likely to develop PTSD following trauma exposure [42].

An additional genetic variation of interest is associated with the dopamine D3 receptor (DRD3) gene. A study showed several SNPs within the DRD3 gene to be associated with PTSD. These SNPs, including rs2134655, rs201252087, rs4646996, and rs9868039, were found to be linked with reduced risk for PTSD. The minor alleles of these SNPs were linked to a decreased likelihood of developing PTSD. In the replication sample of trauma-exposed African American participants, another SNP, rs2251177, showed a nominal association with PTSD in men. The minor allele of this SNP was associated with a lower risk of PTSD [43].

Another relevant variation is found in the dopamine D4 receptor gene (DRD4). DRD4 contains a variable number of tandem repeats (VNTR), with 2 to 10 copies, and the most common variants are 4 and 7 repeats. Numerous studies have investigated the association between DRD4 VNTR and several behavioral and psychiatric conditions, including PTSD. Notably, a significant association has been observed between the DRD4 VNTR and PTSD. Specifically, individuals carrying the 7-repeat allele of DRD4 VNTR have a higher risk of developing PTSD compared to those without this allele [44]. Variations in the solute carrier family 6 member 3 (SLC6A3) gene have also been implicated in the development of PTSD. Specifically, a genetic variation called 3’VNTR9r within the SLC6A3 gene has been associated with lower levels of dopamine reuptake, leading to higher levels of dopamine signaling in certain brain areas. This altered dopamine signaling may affect the brain’s response to stress and trauma, potentially impacting an individual’s ability to cope and increasing their susceptibility to developing PTSD [45].

The dopamine beta-hydroxylase (DBH) gene is another genetic target of interest in PTSD. A single nucleotide polymorphism (SNP) within the DBH gene, rs1611115, has been significantly associated with PTSD. This same variant has also been linked to plasma DBH activity levels, suggesting a potential mechanism through which DBH gene variation may influence susceptibility to PTSD [46]. The catechol-O-methyltransferase (COMT) gene is also involved in dopamine metabolism, and specific genetic polymorphisms within this gene have been associated with PTSD. The functional polymorphism at codon 158, Val158Met (rs4680), affects the amino acid sequence of the COMT enzyme. The Met158 allele of the COMT gene has been associated with an increased risk of developing PTSD and has shown a gene-environment interaction in predicting PTSD when traumatic events are considered. Another COMT gene polymorphism, rs4633C, has also been significantly associated with total PTSD [32,45].

The endocannabinoid system, which involves endogenous cannabinoids and cannabinoid receptors, has also been implicated in PTSD [47]. Alterations in this system, including decreased levels of endocannabinoids like anandamide (AEA) and 2-arachidonoylglycerol (2-AG), have been observed in individuals with PTSD. The endocannabinoid system plays a crucial role in regulating stress and anxiety. Additionally, CB1 receptors, highly expressed in brain regions associated with PTSD, have been implicated in the disorder. Genetic variations in the CB1 receptor gene have been associated with PTSD symptom severity, and preclinical studies suggest that targeting CB1 receptors may hold therapeutic potential for PTSD [47]. The dysregulation of serotonin, norepinephrine, glutamate, GABA, acetylcholine, and dopamine disrupts critical neural circuits and processes underlying emotional regulation, stress response, and cognitive function. Understanding these imbalances and their interplay could pave the way for targeted pharmacological interventions and personalized treatment strategies for individuals affected by PTSD.

### 2.4. Inflammatory System Candidate Gene Variations

Inflammatory system candidate genes are genes that play a role in the body’s inflammatory response. The C-reactive protein (CRP) gene, which encodes the CRP protein involved in the immune system’s response to inflammation, has been associated with PTSD symptoms. Specifically, the SNP rs1130864 within the CRP gene has been significantly linked to increased PTSD symptoms, including hyperarousal symptoms. The genotype of CRP was also associated with the likelihood of a PTSD diagnosis. This SNP was further associated with elevated CRP levels, a protein produced by the liver in response to inflammation. The study found that higher CRP levels were positively correlated with PTSD symptoms and fear-related psychophysiology. These findings suggest that genetic variability in the CRP gene may contribute to the development of PTSD, potentially through an increased pro-inflammatory state [48]. The Interleukin 1 Beta (IL1B) gene, responsible for encoding the interleukin-1 beta protein involved in the immune response, has also been investigated in relation to PTSD. A study found that the minor allele frequencies of IL1B SNPs, rs1143633C, and rs16944A, were significantly lower in patients with PTSD compared to controls. This suggests that these alleles may have a protective effect against PTSD, indicating that the IL1B gene could be involved in the pathogenesis of the disorder [49]. Polymorphisms in the promoter region of the Tumor Necrosis Factor α (TNFα) gene, which encodes the pro-inflammatory cytokine TNFα, have been associated with an increased susceptibility to PTSD. In particular, the SNP rs1800629 in TNFα has been linked to PTSD severity, with the GG genotype identified as a risk genotype for psychiatric disorders. The study also observed significant differences in serum TNFα levels between cases and controls, as well as significant correlations between elevated TNFα levels and PTSD severity [50].

### 2.5. G Protein-Coupled Receptor (GPCR) System Candiate Gene Variations

The Regulator of G-protein signaling-2 (RGS2) gene has been associated with cognitive functioning, particularly in memory and learning processes. Studies have also found an association between RGS2 gene variation and PTSD. For example, one study linked the RGS2 rs4606 allele with PTSD in individuals who experienced a traumatic hurricane under conditions of high stress and low social support [51]. However, further research is needed to understand the specific prosses through which RGS2 gene variations contribute to PTSD. The ADCYAP receptor type I (ADCYAP1R1) gene encodes a G protein-coupled receptor that binds the neuropeptide adenylate cyclase-activating polypeptide (PACAP) and is involved in various physiological processes, including stress response. The ADCYAP1R1 gene has been implicated in stress response, and previous studies have investigated its potential link to PTSD with mixed results. A meta-analysis focusing on the rs2267735 variant in the ADCYAP1R1 gene found that the C allele of this variant was associated with an increased risk for PTSD in the combined sex sample and in the subsample of women and girls, but not in the subsample of men and boys. These findings suggest that the ADCYAP1R1 gene may play a role in PTSD, and there may be sex differences in this association [52]. The Adrenoceptor Beta 2 (ADRB2) gene encodes the beta-2 adrenergic receptor, which is involved in responding to the neurotransmitter norepinephrine and the hormone epinephrine. A study identified an association between a single nucleotide polymorphism (SNP) within the promoter region of the ADRB2 gene, specifically rs2400707, and PTSD symptoms, particularly in interaction with childhood trauma. The rs2400707 polymorphism has been linked to the function of the adrenergic system and was found to be associated with relative resilience to childhood adversity [53]. Further research is necessary to fully uncover the underlying dynamics through which variations in the RGS2, ADCYAP1R1, and ADRB2 genes contribute to PTSD susceptibility and the specific roles of these genetic variations in the development and progression of the disorder.

### 2.6. Other Genes Variations

The brain-derived neurotrophic factor (BDNF) gene, which codes for the BDNF protein, has been extensively studied in relation to PTSD. BDNF is believed to play a role in the development and maintenance of PTSD, and several studies have reported lower levels of BDNF in individuals with PTSD compared to healthy controls. Additionally, lower BDNF levels have been associated with more severe PTSD symptoms. One specific SNP within the BDNF gene that has been investigated is the val66met polymorphism. This SNP leads to a valine (val) to methionine (met) substitution at codon 66 of the BDNF protein. The Met allele of the BDNF Val66Met polymorphism (rs6265) has been associated with reduced levels of BDNF release, heightened HPA axis reactivity, and impaired fear extinction. These factors may contribute to an increased risk for more severe PTSD symptoms [54]. It is important to note that while there is evidence supporting the involvement of the BDNF gene in PTSD, further research is needed to fully understand the mechanisms and implications of these genetic variations in the development and progression of the disorder.

The Monoamine oxidase B (MAOB) gene encodes the MAOB enzyme, which is an isozyme of the MAOA gene and involved in the breakdown of dopamine. The rs1799836 variant in the MAOB gene has been linked to negative emotional personality traits and depression. There is also evidence of a marginally significant association between the MAOB rs1799836 polymorphism and the severity of PTSD symptoms [55]. In addition, the rs1799836 polymorphism has been associated with the severity of PTSD symptoms in male war veterans [44]. The Neuropeptide Y (NPY) gene has been implicated as a potential risk factor for anxiety disorders, including PTSD. A pilot prospective study found that individuals expressing a combination of genetic variants, including the NPY rs16147 polymorphism, were more susceptible to developing PTSD in the absence of early intervention in a high-risk group [56]. The Apolipoprotein E (ApoE) gene has been primarily studied in the context of its association with Alzheimer’s disease. However, recent research suggests that ApoE may also be involved in neuronal and glial responses to stress dysregulation, which is relevant to PTSD. The APOE gene has different isoforms, and the APOE ε4 allele has been linked to the presence of PTSD, particularly re-experiencing/intrusion symptoms [57]. Another study found that combat veterans with the ApoE E2 isoform exhibited higher PTSD scores and alterations in salivary cortisol levels, while mice with the E2 isoform showed impairments in fear extinction and other behavioral, cognitive, and neuroendocrine alterations following trauma [58].

The Oxytocin Receptor (OXTR) gene has been investigated for its role in stress and social behavior, including its potential involvement in PTSD. The rs53576 variant in the OXTR gene has been associated with probable lifetime PTSD [59]. The Fatty Acid Amide Hydrolase (FAAH) gene, which is involved in the regulation of the endocannabinoid system, has been studied in relation to fear learning and PTSD. The FAAH rs324420 polymorphism has been shown to influence physiological, cognitive, and neural signatures of fear learning in women with PTSD [60]. Additionally, the A/A genotype at rs324420 in the FAAH gene was associated with higher PTSD symptom severity and was found exclusively in Black participants [39]. The Protein Phosphatase, Mg^2+^/Mn^2+^ Dependent 1F (PPM1F) gene has been investigated for its involvement in stress response pathways and serotonergic signaling. Variations in the PPM1F gene have been found to moderate the association between PTSD symptom severity and cortical thickness in specific brain regions, including bilateral superior frontal and orbitofrontal regions, as well as the right pars triangularis [61]. The specific SNPs implicated in this study include rs9610608, rs62237483, rs9610645, rs62234965, rs199725385, and rs9610690. The Solute carrier family 18 member A2 (SLC18A2) gene encodes the vesicular monoamine transporter 2 (VMAT2) protein, which is involved in packaging and transporting monoamine neurotransmitters. Several polymorphisms in the SLC18A2 gene have been found in relation to PTSD, with the top SNP being rs363276 [62].

A study investigated the role of the opioid receptor–like 1 (OPRL1) gene in PTSD and fear learning in humans. In particular, the SNP rs6010719, was associated with increased PTSD symptoms in individuals who had experienced moderate to severe child abuse. This association remained significant even after controlling for factors such as age, sex, and substance abuse, which are known to be related to PTSD. The study also found that the association between the G allele carriers of rs6010719 and PTSD risk increased with the degree of trauma exposure. These findings suggest that genetic variations in the OPRL1 gene may contribute to the development and severity of PTSD symptoms [63].

The variant rs4790904 in the protein kinase C alpha (PRKCA) gene has been also associated with PTSD. A study showed that in the Caucasian population, there was a significant correlation between rs4790904 and PTSD symptom clusters (re-experiencing, avoidance and numbing, and hyperarousal). In the African American veterans, a significant association was found between rs4790904 and a current diagnosis of PTSD [64].

The Adenosine Triphosphate (ATP) Synthase Subunit 8 (MT-ATP8) gene encodes a protein subunit of ATP synthase, which is responsible for producing ATP, the main energy currency of the cell. Variations in the MT-ATP8 gene have been associated with PTSD. Specifically, the mt8414C → T variant in this gene has been found to be significantly associated with PTSD [65]. This suggests that mitochondrial genetic variations in the MT-ATP8 gene may play a role in the development of PTSD. The Mitochondrially Encoded NADH Dehydrogenase 5 (MT-ND5) gene is a mitochondrial gene that codes for a subunit of NADH dehydrogenase, which is involved in the electron transport chain within mitochondria. The MT-ND5 gene plays a role in the oxidation of NADH and the transfer of electrons to the respiratory chain. A specific variant in the MT-ND5 gene, mt12501G → A, has been associated with an increased risk of PTSD [65]. These findings suggest that mitochondrial genetic variants, including those in the MT-ND5 gene, may contribute to the development of PTSD and may have implications for the development of treatments targeting mitochondrial function.

The Long Intergenic Non-Protein Coding RNA 1090 (LINC01090) gene, previously known as lincRNA AC068718.1, has been associated with PTSD in women. GWAS identified the rs10170218 variant in the LINC01090 gene as a genetic marker linked to an increased risk of developing PTSD [66]. These findings suggest that non-coding RNA genes, such as LINC01090, may play a role in PTSD.

### 2.7. GWAS Studies

Genome-Wide Association Studies (GWAS) are powerful research approaches used to identify genetic variations associated with complex traits or diseases. These studies involve analyzing the entire genome of individuals to detect genetic markers or variants that are more common in individuals with a particular trait or disease compared to those without it. In the case of PTSD, a variety of associated genes were identified in GWAS. The Retinoid-related Orphan Receptor Alpha (RORA) gene encodes a nuclear hormone receptor that is involved in circadian rhythm regulation and has been implicated in various psychiatric disorders. The association between the rs8042149 variant in the RORA gene and PTSD as shown in GWAS, suggests a potential role of RORA in the development or maintenance of PTSD [67]. Further research is needed to elucidate the specific mechanisms by which RORA may contribute to the development of PTSD. In a similar manner, the Phosphoribosyl Transferase Domain Containing 1 (PRTFDC1) gene has been associated with PTSD. Specifically, the rs6482463 variant in the PRTFDC1 gene has been linked to PTSD [68]. The exact function of PRTFDC1 in relation to PTSD is not well understood, and further studies are required to explore its role and potential mechanisms.

Similarly, the Tolloid-Like 1 (TLL1) gene has been identified as a susceptibility gene for PTSD. Specifically, the rs6812849 and rs7691872 variants in the TLL1 gene have shown replicated associations with PTSD, and rs406001 reached genome-wide significance in the initial GWAS [69]. These findings suggest that TLL1 may play a role in the development or vulnerability to PTSD. The Ankyrin Repeat Domain 55 (ANKRD55) gene, located on chromosome 5, has been implicated in PTSD susceptibility. A genome wide significant association was found for the rs159572 variant in the ANKRD55 gene [70]. In a related framework, the Zinc finger protein 626 (ZNF626) gene has shown a genetic association with PTSD. Specifically, the rs11085374 variant in the ZNF626 gene showed a suggestive association with PTSD [70]. All this information is summarized below as shown in Table 1.

**Table 1 biology-12-01050-t001:** Genetic variations represented as alterations or polymorphisms in specific genes involved in key biological processes relevant to the pathogenesis of PTSD and identified as catalytic factors associated with the disorder.

Type	Gene	Variation	Ref.
Hypothalamic-pituitary-adrenal axis	*NR3C1*	Bcl-1 (rs41423247) ER22/23EK (rs6189/rs6190) 9β SNP (rs6198) rs258747	[17] [21] [21] [22]
	*FKBP5*	rs3800373 rs9296158 rs1360780 rs9470080	[22] [22] [22] [22]
	*CRHR1*	rs110402 rs12938031 rs4792887 rs12944712	[24] [24] [25] [25]
	*CRHR2*	rs2267715	[26]
Serotonergic system	*SLC6A4*	5-HTTLPR/rs25531 rs16965628	[30] [70]
	*HTR2A*	rs7997012 rs977003 rs7322347	[5] [5] [5]
	*TPH2*	rs11178997T	[32]
Dopaminergic system	*DRD2/ANKK1*	DRD2/ANKK1-Taq1A (rs1800497)	[71]
	*DRD3*	rs2134655 rs201252087 rs4646996 rs9868039 rs2251177	[72] [72] [72] [72] [72]
	*DRD4*	VNTR 7R	[44]
	*SLC6A3*	3′VNTR9r	[45]
	*DBH*	rs1611115	[46]
	*COMT*	rs4680 rs4633C	[44] [32]
GABAergic ^1^ system	*GABRA2*	Unspecified	[36]
Endocannabinoid system	*SKA2*	rs7208505	[37]
	*CNR1*	G1359A (rs1049353) rs806371	[38] [39]
Inflammatory system	*CRP*	rs1130864	[48]
	*IL1B*	rs1143633*C* and *rs16944*A	[49]
	*TNFα*	rs1800629	[50]
GPCR ^2^ system	*RGS2*	rs4606	[51]
	*ADCYAP1*	rs2267735	[52]
	*ADRB2*	rs2400707	[53]
Other	*MAOB*	rs1799836	[44]
	*NPY*	rs16147	[56]
	*APOE*	ε4 ε2	[57] [58]
	*OXTR*	rs53576	[59]
	*FAAH*	rs324420	[39]
	*PPM1F*	rs9610608	[61]
	*SLC18A2*	rs363276	[62]
	*OPRL1*	rs6010719	[63]
	*PRKCA*	rs4790904	[64]
Brain-derived neurotrophic factor	*BDNF*	val66met (rs6265)	[73]
Mitochondrial	*MT-ATP8*	mt8414C → T	[65]
	*MT-ND5*	mt12501G → A	[65]
RNA regulation	*LINC01090*	rs10170218	[66]
GWAS ^3^ studies	*RORA*	rs8042149	[74]
	*PRTFDC1*	rs6482463	[68]
	*TLL1*	rs6812849 rs7691872 rs406001	[69] [69] [69]
	*ANKRD55*	rs159572	[70]
	*ZNF626*	rs11085374	[70]

^1^ Gamma-Aminobutyric Acid (GABA)-ergic; 2 G-protein coupled receptor; 3 Genome-Wide Association Study.

## 3. Exploring the Molecular Landscape: Gene Ontology and Network Analyses of Key PTSD-Associated Genes

In this section, a comprehensive analysis is undertaken into the molecular landscape of PTSD by conducting gene ontology and network analyses focused on the main genes associated with the disorder as provided in the table above (Table 1). Gene ontology analysis allows us to explore the functional annotations and categorizations of these genes, providing insights into the specific molecular processes and pathways they are involved in. By unravelling their roles within these ontological frameworks, we gain a deeper understanding of how these genes contribute to the development and manifestation of PTSD. Additionally, we employ network analyses to investigate the interconnectedness and associations among the identified PTSD-associated genes. By constructing biological networks, we can explore the intricate relationships and potential interactions between these genes. This network perspective allows us to uncover novel connections and identify key hubs or modules within the molecular network that might play critical roles in PTSD pathophysiology. Through gene ontology and network analyses, we aim to examine the functional implications and molecular mechanisms underlying the identified PTSD-associated genes. This comprehensive exploration provides valuable insights into the intricate network of interactions and pathways involved in PTSD, facilitating a deeper understanding of the disorder and potentially identifying new therapeutic targets. Ontological analysis involves the study of biological data, such as genes, by categorizing them based on their functional properties and relationships. This analysis relies on ontologies, that organize biological concepts into a hierarchical framework. Gene annotation cluster networks, identify groups or clusters of genes with similar functional annotations which are constructed by examining the shared annotations or functional characteristics of genes. GeneCodis4 was utilized to identify significant associations between genes based on their shared annotations.

### 3.1. Biological Processes

The analysis of gene categories based on Gene Ontology (GO) biological processes (BP) using GeneCodis4 provides valuable insights into the potential roles of specific genes in various biological functions related to PTSD and can be seen in Table 2. The corresponding gene-annotation cluster network is shown in Figure 1. The identified genes are associated with diverse processes that contribute to the development and manifestation of PTSD. One prominent category is the response to amphetamine, which involves genes such as OXTR, DBH, DRD4, SLC18A2, and RGS2. Amphetamines can influence neurotransmitter systems and have been linked to the regulation of dopamine and norepinephrine [75]. The involvement of these genes suggests a potential role in modulating the response to amphetamines and their impact on PTSD symptoms. Another significant category is memory, which encompasses genes such as OXTR, BDNF, DBH, CNR1, HTR2A, and SLC6A4. Memory processes play a crucial role in PTSD, as individuals often experience intrusive memories and flashbacks related to the traumatic event [76]. The genes associated with memory indicate their involvement in the formation, consolidation, and retrieval of traumatic memories, providing potential targets for therapeutic interventions. As it was previously mentioned the GABA system and genes involved in GABAergic neurotransmission are also implicated in PTSD. GABA is an inhibitory neurotransmitter that plays a crucial role in regulating neuronal excitability. Understanding the specific genes involved in the GABA system can offer further clarifications on the mechanisms underlying GABAergic dysfunction in PTSD and guide the development of targeted treatments. The identified genes related to fear response, social behavior, and behavioral response to substances like ethanol and cocaine highlight the complex interactions between genetic factors and behavioral manifestations of PTSD. These genes, such as CRHR1, DRD4, OXTR, and SLC6A3, provide insights into the molecular pathways involved in fear conditioning [77], social behavior [78], and substance abuse [79], which are often observed in individuals with PTSD.

### 3.2. Cellular Components (CC)

By exploring the enriched cellular components of the identified PTSD-associated genes, as revealed by Gene Ontology (GO) Cellular Components (CC) analysis, the aim is to gain insights into the biological processes and systems involved in PTSD. The main findings of this analysis can be seen in Table 3 and the corresponding gene-annotation cluster network in Figure 2. As expected, based on the gene information provided in previous sections, the most enriched cellular components include neurotransmitter systems. The involvement of genes related to dopaminergic, glutamatergic, and GABA-ergic synapses suggests perturbations in these neurotransmitter systems in PTSD. Disruptions in dopamine signaling (SLC6A3, DRD3) and glutamatergic transmission (APOE, HTR2A) have been linked to emotional regulation and fear responses, while GABA-ergic dysfunctions (GABRA2, CNR1) may influence anxiety and stress-related behaviors. Other noteworthy findings include the enrichment of genes involved in neuron projection and dendrites which suggest their crucial role in PTSD [80]. These components are associated with neuronal connectivity, information processing, and signal transmission. The involvement of genes like SLC6A3, TPH2, GABRA2, and HTR2A indicates potential disruptions in neurotransmitter systems, including serotonin, dopamine, and GABA, which have been implicated in mood regulation and stress responses. The integral components of presynaptic and postsynaptic membranes, as well as synapses in general, highlight the importance of synaptic signaling in PTSD [81]. Genes like SLC6A4, CNR1, and HTR2A, which are enriched in these components, are involved in modulating neurotransmission and synaptic plasticity. Dysregulation within these synaptic networks may contribute to altered information processing and the development of PTSD symptoms. The enrichment of genes in plasma membrane components suggests alterations in cell surface receptor systems involved in signaling and information transfer. Genes such as ADRB2, CRHR1, and OXTR indicate potential involvement of adrenergic and neuropeptide systems in PTSD. These systems are shown to play a role in stress response regulation and emotional processing [82].

### 3.3. Molecular Functions (MF)

The Gene Ontology (GO) Molecular Functions (MF) table below (Table 4) and the relevant gene-annotation cluster network (Figure 3) provide valuable perspectives into the enriched gene activities associated with specific functional categories. The significant enrichment of genes involved in monoamine transmembrane transporter activity, such as SLC6A3, SLC6A4, and SLC18A2, highlight the importance of these transporters in molecular signaling within the nervous system. Monoamine neurotransmitters, including dopamine, serotonin, and norepinephrine, play essential roles in regulating mood, emotion, and cognition. The activity of these transporters is crucial for the reuptake and recycling of monoamines, ensuring their proper function and maintaining neurotransmitter balance. Dysregulation of these transporters has been previously implicated in PTSD [83]. Unsurprisingly, based on information provided above the enrichment of genes associated with GPCR activity suggests the involvement of GPCR-mediated signaling pathways in the molecular processes underlying PTSD. The identified genes participate in modulating neurotransmitter systems, stress responses, and other physiological functions. As expected, the enrichment of CRH receptor activity, represented by CRHR1 and CRHR2, as well as the enrichment of dopamine neurotransmitter receptor activity, particularly through Gi/Go signaling, indicates their involvement in the molecular processes underlying PTSD. The enrichment of genes involved in serotonin:sodium symporter activity (SLC6A4, SLC18A2), in serotonin binding (HTR2A, SLC6A4), and in peptide hormone binding (CRHR1, OXTR, CRHR2) once again highlight the potential impact of serotonin and peptide hormones in the pathophysiology of PTSD. Peptide hormones like CRH and oxytocin play critical roles in stress responses, social behavior, and emotional regulation.

## 4. Proposed Function for Catalytic Factors

The deterministic model of the stimulus/traumatic event is described by a partial differential equation of the form:
(1)$$\partial T/\partial t=\;▽\left(k\left(T\right)▽T\right)+\lambda T^{P}$$ where T is the stimulus/traumatic event and is dependent on both time (t) and spatial coordinates $\vec{x}=\left(f_{1},f_{2},f_{3}\right)$. $f_{1},f_{2},f_{3}$ correspond to genetic factors, molecular factors, and epigenetic factors respectively. Each factor contributes to the overall response of the system to the traumatic event and has corresponding weights on the system’s behavior. ∂T/∂t represents the rate of change of the stimulus/traumatic event *T* with respect to time. The term $▽\left(k\left(T\right)▽T\right)$ represents the diffusion of the traumatic event, where ▽T represents the traumatic event gradient and $k\left(T\right)$ represents the traumatic event diffusion coefficient. The term $\lambda T^{P}$ represents the catalytic effect, where λ represents the catalytic constant and *P*, where 0 < *P* < 2, represents the power law exponent. In the context of this equation, it determines the relationship between the traumatic stimulus (*T*) and the likelihood of developing PTSD. More in particular, the focus will be on the study of the steady state of the above partial differential equation since it will determine the areas of stability in the bifurcation diagram.

Overall, this partial differential equation provides a mathematical framework for understanding the dynamics of the stimulus/traumatic event and its interactions with genetic, molecular, and epigenetic factors. These characteristics were described by functions with the aim to incorporate them into a holistic mathematical model for PTSD.

## 5. Conclusions

The investigation of catalytic factors in the context of PTSD offers a promising avenue of research with significant potential to advance our understanding of this complex disorder. PTSD is a multifactorial disorder influenced by genetic, molecular, and epigenetic factors. This work focused solely on the genetic factors associated with PTSD. Future work will also examine the epigenetic and molecular factors. The categorization of genes based on gene ontology and the identification of enriched gene activities provide critical insights into the underlying molecular processes and cellular components involved in PTSD pathogenesis. This knowledge forms the foundation for targeted interventions that focus on specific cellular constituents, such as neurotransmitter systems, synaptic function, and cell surface receptors. The presented partial differential equation offers a comprehensive mathematical framework for studying the dynamics of the stimulus in the context of PTSD. By incorporating genetic, molecular, and epigenetic factors through their corresponding weights, the function captures the complex interplay between these factors and the stimulus. The diffusion term accounts for the spread of the traumatic event, while the catalytic effect, determined by the catalytic constant (*λ*) and power law exponent (*P*), influences the relationship between the stimulus and the likelihood of developing PTSD. The study of the steady state of this equation enables the identification of stable regions in the bifurcation diagram, providing valuable insights into the system’s behavior. This mathematical framework lays the foundation for a holistic model of PTSD, integrating diverse factors and facilitating a deeper understanding of the disorder. Moreover, the development of comprehensive databases will facilitate data integration and enable a deeper understanding of the disorder’s complexity. This research will undoubtedly contribute to a profound understanding of the intricate mechanisms underlying PTSD and, subsequently, facilitate the development of more efficacious prevention and treatment strategies. By utilizing the aforementioned approaches, the potential for personalized and evidence-based interventions tailored to the unique requirements of individuals afflicted with PTSD becomes an attainable aim. By integrating knowledge from diverse disciplines and considering the multifaceted nature of the disorder, the field is poised to advance the frontier of PTSD research and translate findings into practical applications that enhance patient outcomes.

## Figures and Tables

**Figure 1 biology-12-01050-f001:**
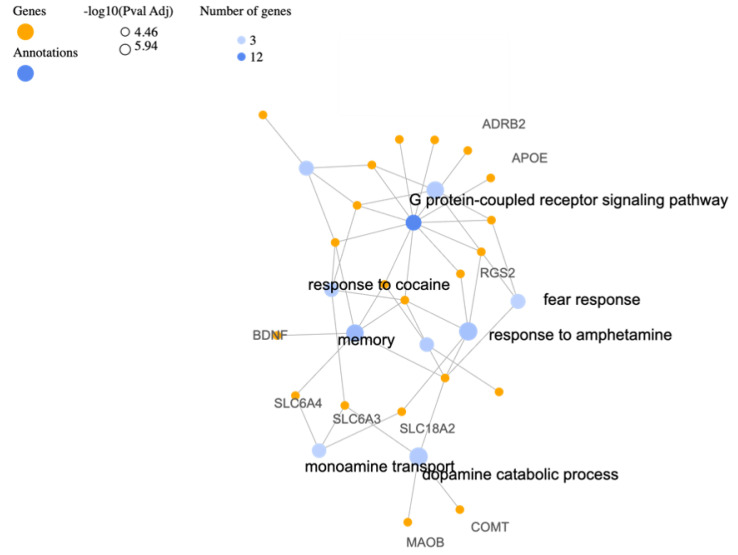
Gene-annotation cluster network following visualizations generated for the top 10 terms of related categories with the identified PTSD-associated gene list for GO biological process.

**Figure 2 biology-12-01050-f002:**
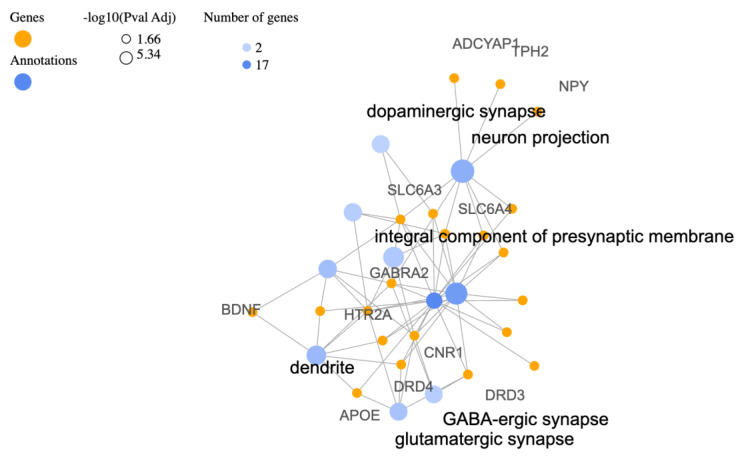
Gene−annotation cluster network following visualizations generated for the top 10 terms of related categories with the identified PTSD−associated gene list for GO cellular components.

**Figure 3 biology-12-01050-f003:**
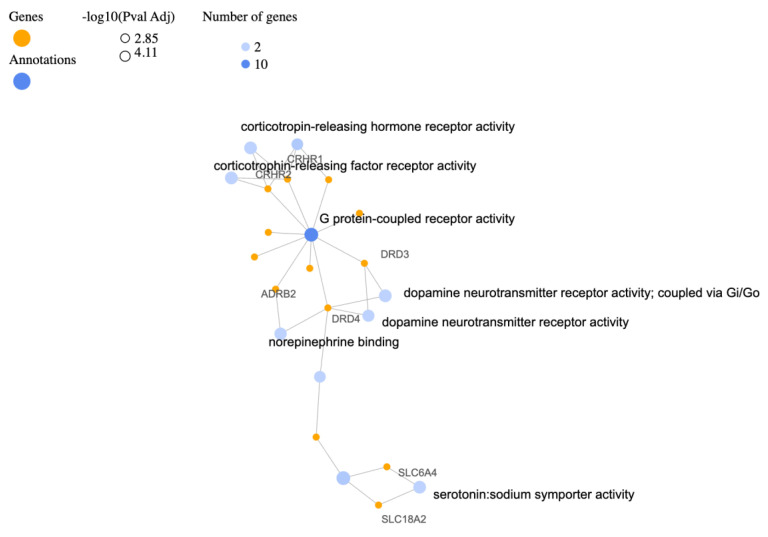
Gene−annotation cluster network following visualizations generated for the top 10 terms of related categories with the identified PTSD−associated gene list for GO molecular functions.

**Table 2 biology-12-01050-t002:** Top 15 enriched biological processes are represented as activities or pathways in which the proteins encoded by the PTSD-associated genes are known to participate.

Description	Relative Enrichment	Genes
response to amphetamine	82.7	OXTR, DBH, DRD4, SLC18A2, RGS2
dopamine catabolic process	211.8	SLC6A3, MAOB, DBH, COMT
memory	36.5	OXTR, BDNF, DBH, CNR1, HTR2A, SLC6A4
negative regulation of voltage-gated calcium channel activity	141.2	CRHR1, OPRL1, DRD3, DRD4
G protein-coupled receptor signaling pathway	6.7	ADRB2, CRHR1, OXTR, OPRL1, APOE, DRD3, CNR1, ADCYAP1, HTR2A, CRHR2, DRD4, RGS2
fear response	226.9	CRHR1, DBH, DRD4
negative regulation of blood pressure	75.6	NPY, OPRL1, DRD3, CNR1
positive regulation of vasoconstriction	66.2	FAAH, OXTR, DBH, HTR2A
response to cocaine	62.3	SLC6A3, OXTR, DRD3, CNR1
monoamine transport	176.5	SLC6A3, SLC6A4, SLC18A2
behavioral response to ethanol	158.9	CRHR1, DBH, DRD4
response to ethanol	25.7	SLC6A3, MAOB, DRD3, CNR1, RGS2
behavioral response to cocaine	113.5	DRD3, HTR2A, DRD4
social behavior	40.7	OXTR, DRD3, DRD4, SLC6A4
dopamine metabolic process	99.3	DRD3, DRD4, COMT

**Table 3 biology-12-01050-t003:** Top 15 enriched cellular components represented as specific structures or compartments within the cell where the identified genes implicated in PTSD are known to play important roles.

Description	Relative Enrichment	Genes
neuron projection	11.9	SLC6A3, TPH2, CRHR1, NPY, OPRL1, GABRA2, ADCYAP1, SLC6A4, RGS2
integral component of plasma membrane	5.2	ADRB2, SLC6A3, CRHR1, OXTR, OPRL1, GABRA2, DRD3, CNR1, HTR2A, CRHR2, DRD4, SLC6A4, SLC18A2
integral component of presynaptic membrane	41.7	SLC6A3, CNR1, HTR2A, SLC6A4
dendrite	8.5	BDNF, GABRA2, APOE, HTR2A, CRHR2, DRD4, COMT
axon	8.5	SLC6A3, BDNF, GABRA2, CNR1, HTR2A, COMT
integral component of postsynaptic membrane	32.4	SLC6A3, HTR2A, SLC6A4
dopaminergic synapse	104.1	SLC6A3, SLC18A2
glutamatergic synapse	8.9	APOE, DRD3, CNR1, HTR2A, DRD4
GABA-ergic synapse	24.9	GABRA2, DRD3, CNR1
plasma membrane	1.9	ADRB2, PRKCA, SLC6A3, CRHR1, OXTR, OPRL1, GABRA2, APOE, DRD3, CNR1, HTR2A, CRHR2, DRD4, SLC6A4, SLC18A2, RGS2, COMT
synapse	4.2	DBH, GABRA2, DRD3, CNR1, HTR2A, SLC6A4
presynapse	10.9	CNR1, HTR2A, SLC6A4
discoidal high-density lipoprotein particle	286.1	APOE
serotonergic synapse	286.1	SLC6A4
cytoplasmic vesicle	4.1	NPY, OPRL1, DBH, GABRA2, HTR2A, SLC18A2

**Table 4 biology-12-01050-t004:** Top 15 enriched molecular functions represented as specific activities or biochemical processes that the proteins encoded by the PTSD-associated genes are known to perform.

Description	Relative Enrichment	Genes
monoamine transmembrane transporter activity	172.9	SLC6A3, SLC6A4, SLC18A2
G protein-coupled receptor activity	7.1	ADRB2, CRHR1, OXTR, NPY, OPRL1, DRD3, CNR1, HTR2A, CRHR2, DRD4
corticotropin-releasing hormone receptor activity	345.9	CRHR1, CRHR2
dopamine neurotransmitter receptor activity, coupled via Gi/Go	345.9	DRD3, DRD4
corticotrophin-releasing factor receptor activity	345.9	CRHR1, CRHR2
serotonin:sodium symporter activity	345.9	SLC6A4, SLC18A2
norepinephrine binding	259.5	ADRB2, DRD4
dopamine neurotransmitter receptor activity	207.6	DRD3, DRD4
peptide hormone binding	36.2	CRHR1, OXTR, CRHR2
dopamine binding	148.3	SLC6A3, DRD4
serotonin binding	86.5	HTR2A, SLC6A4
neurotransmitter transmembrane transporter activity	57.7	SLC6A3, SLC6A4
neurotransmitter receptor activity	18.8	GABRA2, HTR2A, DRD4
signaling receptor binding	6.1	SLC6A3, BDNF, NPY, APOE, ADCYAP1
G protein-coupled serotonin receptor activity	30.6	HTR2A, DRD4

## Data Availability

Not applicable.

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
