# Peer review of "Assessing and Modelling of Post-Traumatic Stress Disorder Using Molecular and Functional Biomarkers"

_biology, 2023, doi:10.3390/biology12081050_

Round 1

Reviewer 1 Report

This is a comprehensive account of molecular biomarkers underlying post traumatic stress disorder. The authors present a review of catalytic factors known to be associated with PTSD by citing various studies that have identified genes and SNPs associated with this disorder. They outline the difficulty in classifying these genes as being consistently up or downregulated these given the complexity of the nature of the disorder and heterogeneity of the affected individuals. Additionally, they perform network analysis on these by conducting GO analysis. Finally, they propose a modelling paradigm using linear differential equations.

I think the manuscript is well written, but a little dense. I have a few minor comments that I would like to be addressed

- I would recommend that the authors cut down on some details (e.g. of specific SNPs) in the text and instead referencing a table / supplementary table wherein these SNPs are summarized and perhaps colour coded to indicate whether they are associated with high or low expressed of that gene

- I am curious about what gene-set the authors used as the "background" set for their GO analysis

- Citation may be missing for line 138: In individuals with PTSD, alterations in epigenetic regulation of GR gene expression and GC signaling pathway genes have been reported

- It would be good for the authors to estimate a few different parameters and plot the modelled molecular pathway and proposed theoretical attenuation of symptoms given therapeutic intervention

I am not sure what the authors mean by reduction in GABAergic "tone" and I think it might be confusing to the general reader to encounter this word

I would consider rewording section 2.2 title "HPA Axis and related genes variations" 

The sentence should be changed (lines 55-57):

Understanding the role of catalytic factors in 55 PTSD can lead to valuable insights regarding the heterogeneity of symptom presentation 56 and treatment response among individuals affected by PTSD are gained. 

Author Response

- I would recommend that the authors cut down on some details (e.g. of specific SNPs) in the text and instead referencing a table / supplementary table wherein these SNPs are summarized and perhaps colour coded to indicate whether they are associated with high or low expressed of that gene

Thank you for your valuable feedback on our manuscript. We believe that since the SNPs are shown in the table, they should also be mentioned at some point in the text. Our aim was to create a complete table with all genetic variations associated in PTSD. The focus was on SNPs and not gene expression, seeing as if they are present they count as a catalytic factor regardless of whether the gene shows high-or low- expression. However, this is a great idea that we will incorporate in future work.

- I am curious about what gene-set the authors used as the "background" set for their GO analysis

Lines 531-533. The gene-set used is the one from table 1.

- Citation may be missing for line 138: In individuals with PTSD, alterations in epigenetic regulation of GR gene expression and GC signaling pathway genes have been reported

The citations have been included in the rest of the paragraph. A small change in this sentence has been made to indicate this.

- It would be good for the authors to estimate a few different parameters and plot the modelled molecular pathway and proposed theoretical attenuation of symptoms given therapeutic intervention

The equation has been adjusted to include molecular catalytic factors, genetic catalytic factors and epigenetic catalytic factors. The focus of this work is on the genetic catalytic factors (solely due to the size of the information that needs to be presented). The partial differential equation shown is a mathematical framework for understanding the dynamics of the stimulus/traumatic event and its interactions with genetic, molecular, and epigenetic factors. The aim is to look at PTSD from the side of the stimulus and the cascade of events that follow that may lead to PTSD development.

- I am not sure what the authors mean by reduction in GABAergic "tone" and I think it might be confusing to the general reader to encounter this word

Changed to “overall activity and functioning” in line 246.

- I would consider rewording section 2.2 title "HPA Axis and related genes variations" 

Done.

- The sentence should be changed (lines 55-57):

Understanding the role of catalytic factors in 55 PTSD can lead to valuable insights regarding the heterogeneity of symptom presentation 56 and treatment response among individuals affected by PTSD are gained

Done.

We would like to thank again Reviewer for his/her positive comments.

Reviewer 2 Report

The authors delve deeply into different factors that may influence or may be involved in the complex PTSD disorder. Some factors the authors detail on include different imbalanced neurological pathways as well as genetic variations that may act as catalysts to this dysregulation. This is a timely report which I have read with interest. I have a main comment that may help in improving the manuscript.

  - The review is dominant on gene variations. Please restructure section 2.6 'Other gene variations' into more concise paragraphs, such as neuropeptide and neurotrophic gene variations etc.   If needed, the authors may reduce the information provided or try to structure it using a figure/table. The content of the paragraph should be reflected in the title of that section. For instance, BDNF is significant enough to warrant a paragraph on its own. The NPY part is under-described. Instead of quantity, it is best to focus on a shorter list that is more thoroughly explained.

Author Response

  - The review is dominant on gene variations. Please restructure section 2.6 'Other gene variations' into more concise paragraphs, such as neuropeptide and neurotrophic gene variations etc. If needed, the authors may reduce the information provided or try to structure it using a figure/table. The content of the paragraph should be reflected in the title of that section. For instance, BDNF is significant enough to warrant a paragraph on its own. The NPY part is under-described. Instead of quantity, it is best to focus on a shorter list that is more thoroughly explained.

Thank you for your comments and suggestions. Seeing as there are a lot of genes and SNPs included in the paragraph ‘Other gene variation’ we believe it works best if we keep it together rather than create 7 separate paragraphs. For that reason, BDNF is included in this paragraph but has its own row in the corresponding table. The aim of the paper is to identify all genetic variations associated with PTSD that will be used in accordance with the proposed model in the future. Therefore, we focused on creating a thorough complete list and not explaining in great detail the associated SNPs that can be found in other studies as references. Lastly, the amount of information provided for each gene is in accordance with the relevant SNPs.

Reviewer 3 Report

The manuscript entitled,Assessing… needs revision before it could be considered for publication. Some comments below.

Major: What is your rationale for why did you come up with this review? It would help if you strengthen this. Major neurologic target pathways or diagram for PTSD target is needed here for clarity.

Comments:

1.       Line 21: What is nature here?

2.       Grammar should be checked.

3.       For 2.1. Make a flow or figure for biomarkers of oxidative stress.

4.       Line 194. Imbalances should be imbalances; please also check others.

5.       2.3. Tabulate neurotransmitters and genes associated with PTSD, or transfer Table 1 here. Make also Table 1 more informative.

6.       Line 531. What is delve? Make it more scientific.

7.       Make all tables more informative and organized.

8.       Strengthen your conclusion.

Make it more scientific

Author Response

Major: What is your rationale for why did you come up with this review? It would help if you strengthen this. Major neurologic target pathways or diagram for PTSD target is needed here for clarity.

We would like to thank Reviewer for his/her positive comments. As mentioned in the paper, the future aim is to develop a holistic model of PTSD that will incorporate a panel of characteristics. One of these characteristics is the catalytic factors associated with PTSD and more in particular the genetic, molecular, and epigenetic catalytic factors. This paper focused on identifying the genetic catalytic factors and proposing the corresponding function that in the future will be incorporated in the mathematical model for PTSD. This paper doesn’t aim in examining PTSD from a neurologic POV but study the traumatic event that may lead to PTSD from a mathematical POV.

Comments:

  1. Line 21: What is nature here?

Changed to “associations”.

  1. Grammar should be checked.

Done.

  1. For 2.1. Make a flow or figure for biomarkers of oxidative stress.

This paper only focuses on the genetic catalytic factors associated with PTSD.

  1. Line 194. Imbalances should be imbalances; please also check others.

According to MDPI guidelines, all subheadings should be in capital letters and the corresponding changes have been made.

  1. 2.3. Tabulate neurotransmitters and genes associated with PTSD, or transfer Table 1 here. Make also Table 1 more informative.

Table 1 is provided at the end of all the paragraphs that mention genes and SNPs associated with PTSD thus is was inserted after 2.7.

  1. Line 531. What is delve? Make it more scientific.

Rephrased.

  1. Make all tables more informative and organized.

Done.

  1. Strengthen your conclusion.

Done

Comments on the Quality of English Language

Make it more scientific

Done

 We revised the manuscript and tables and reorganized last section enriching our proposed model. We hope this clarification provides a better understanding. Thank you very much for your valuable feedback.